# Use of Shared-Mobility Services to Accomplish Emergency Evacuation in Urban Areas via Reduction in Intermediate Trips—Case Study in Xi'an, China

**Menghui Li [1], Jinliang Xu [1,\*], Xingliang Liu [1], Chao Sun [2] and Zhihao Duan [1]**

[1] College of Highway Engineering, Chang'an University, Xi'an 710064, China;
menghui_li@outlook.com (M.L.); xingliang1125@outlook.com (X.L.); zhihao.duan@chd.edu.cn (Z.D.)

[2] School of Automotive and Traffic Engineering, Jiangsu University, Zhenjiang 212013, China;
chaosun@ujs.edu.cn

\* Correspondence: xujinliang@chd.edu.cn

**Abstract:** Under no-notice evacuation scenarios with limited time horizons, the effectiveness of evacuation can be negatively impacted by intermediate trips that are made by family members and the identification of vulnerable populations. The emergence of shared-mobility companies, such as Uber and DiDi, can be considered as a potential means to address above-mentioned concerns. The proposed study explores the utility of shared-mobility services under emergency-evacuation scenarios and makes recommendations to relevant bodies that are based on the obtained and they are discussed herein. The study investigates attitudes of the public, experts, and drivers towards the use of shared-mobility resources during emergency evacuations based on a stated preference survey. Results of questionnaires, driver interviews, and face-to-face expert interviews have been analyzed to validate the feasibility and identify potential problems of leveraging shared-mobility services during evacuation response, especially in metropolitan areas wherein such services are already ubiquitous. Numerical simulations have been performed to quantify potential improvements in the total trip distance and number of evacuees after incorporating the use of shared mobility into emergency-response operations. However, despite the observed improvement in emergency efficiency, certain realistic roadblocks must be overcome. Realization of the proposed objective heavily depends on actionable policy recommendations, provided herein as a reference for the government, emergency management agencies, and shared-mobility companies.

**Keywords:** shared mobility; emergency evacuation; intermediate trips; vulnerable population

## 1. Introduction

Evacuation planning should be regarded as a key part of emergency planning [1]. The Sendai Framework for Disaster Risk Reduction [2] indicates four priorities for action, where the fourth priority focused on disaster preparedness for effective response. Evacuation is the main macro-activity to reduce exposure, one of the three dimensions of disaster risk [3]. Human behavior during emergency evacuations is an important consideration with regard to disaster-response procedures. Of the many challenges that exist in this regard, two aspects have attracted increased attention in recent years. The first is the effect of the number of intermediate trips required to be made during an emergency evacuation [4–6], and the other concerns the evacuation of vulnerable populations [7–9]. The term "intermediate trips" refers to journeys that are undertaken by evacuees that do not end at the envisioned destination [10]. For instance, evacuation of private vehicles often involves undertaking a number of small trips—to pick up family members—that ultimately constitute trip chains leading up to the

final destination [5,11]. Vulnerable populations, on the other hand, refer to groups of people in need of evacuation assistance [4], such as those with no access to private vehicles, the elderly, low-income groups, language-disadvantaged groups, the physically and mentally disabled, and children/youths [12–14].

Gathering of family members is a widely observed phenomenon during emergencies [15–17]; consequently, intermediate trips are inevitably required to be made during no-notice evacuations. The said intermediate trips, however, are seldom considered during evacuation modeling performed by transportation researchers. Most modeling algorithms assume that evacuees undertake a single trip, and hence, ignore the fact that an evacuation trip might, in fact, be part of a chain [17]. This conflict between actual evacuee behavior and evacuation modeling, which tends to overlook actions that are observed by social scientists, in turn, affects the clearance time of the evacuation network [16,18]. Compared to evacuation models that do not consider intermediate trips, significantly longer network clearance times can be expected once the impact of gathering behavior is incorporated [17]. Furthermore, certain traffic strategies (left-turn elimination, contraflows, etc.) tend to be implemented at inappropriate times, thereby tending to impede people performing the said intermediate trips.

Evacuation of vulnerable populations has previously been highlighted after the occurrence of hurricanes Katrina and Rita, due to which mass casualties were encountered owing to the lack of evacuation planning for the affected population. Among the 1500 and more casualties that were caused by the hurricane Katrina, most belonged to the vulnerable group sans access to private vehicles [19]. A similar phenomenon trend was observed during evacuation post occurrence of the Fukushima nuclear power-plant disaster in Japan. The authors believe that the most feasible method to evacuate vulnerable populations involves organizing mass transportation systems to serve the locals during emergencies. Over the past decade, a number of researches have investigated transit-based and multimodal evacuation techniques that are based on the optimization of scheduling and operations [20–23]. Along with research concerning transit-based evacuation of the vulnerable population, studies focusing on persons with disabilities [24], availability of hospitals and special facilities [25,26], as well as those concerning the elderly [27,28] have drawn less attention in recent years in view of the challenges—lack of personnel, monetary resources, engagement, etc.—associated with their solutions. The biggest problem facing the evacuation of vulnerable populations involves identifying locations of these people, many of whom are not recognized by official systems.

Shared mobility refers to an emerging transport service that connects registered drivers with passengers via mobile devices and applications [29]. The emergence of shared-mobility companies, such as Uber and DiDi, offers a feasible means to overcome above-mentioned issues. As the concept of the sharing economy becomes increasingly prevalent, the "shared" lifestyle is being widely accepted among people. The first use of shared-mobility services for emergency evacuation can be traced back to hurricane Sandy, which occurred in 2012. Thenceforth, shared-mobility services have been utilized on several occasions under different emergency situations, such as the winter storm Nemo in 2013, winter storm Juno and Houston floods in 2015, Louisiana floods and hurricane Matthew in 2016, and so on. Although the shared-mobility service has faced its fair share of problems, such as price gouging [30], it has proved its potential with regards to providing pressure relief for emergency management agencies (EMAs) and the availability to provide service during evacuation up to a certain degree. Recently, Clewlow and Laberteaux [31] found that shared mobility were reaching new markets unmet by traditional carsharing services based on the results of a comprehensive travel and residential survey deployed in five major United States (U.S.) cities that included questions on the adoption and use of carsharing and shared-mobility services. Alemi et al. [32] investigate the factors affecting the adoption of shared mobility among population born between 1965 to 1997 in California. A series factors have been found related to the usage of these services. Their findings provide a starting point for efforts to forecast the adoption of shared mobility and their impacts on overall travel patterns across various regions and sociodemographics. A systematic review of shared mobility was presented by Jin et al. [33], which was mainly focused on its impact on the equity, efficiency, and sustainability of

urban development. Though a series researches have been proceeded to link shared mobility to our current knowledge, however, to the best of the authors' knowledge, no relevant study has been performed to date to explore the use of shared-mobility services as a substitute for intermediate trips and as a means to identify vulnerable populations during emergency-evacuation scenarios. In the event of emergency situations, government bodies utilize dedicated communication channels in partnership with local telecommunication providers and establish procedures to initiate evacuations. However, this information is not available to the public. When compared to evacuation plans and emergency communication channels, which the general population would never use if no disaster occurred, people are more accustomed to using common mobile-phone applications on a daily basis. The shared-mobility platform can, therefore, act as the first mode of receiving information as well as reorganizing and delivering the same to governmental agencies. Each driver participating in the evacuation program can be considered as an information receiver; this number would far exceed that of telephone operators hired by EMAs. Consequently, there exists a great need for relevant research to be undertaken concerning the use of shared-mobility services in the event of emergency evacuations. In this study, shared mobility acts as a disaster risk reduction (DDR) initiative, with its potential of scalability being influenced by support from national, regional and/or local authorities [34]. Among the four components of evacuation planning (mitigation, preparedness, response, recovery), shared mobility would play a role in "response" if well organized during "preparedness", as "preparedness" is essential to verify feasibility of shared-mobility resources during emergency evacuations.

The proposed study explores the potential for leveraging shared-mobility services to eliminate the need for making intermediate trips during emergency evacuations and introduces shared mobility as an effective method for evacuating vulnerable populations through the establishment of public–private partnerships. To facilitate the establishment of such partnerships and achieve the said objective, the authors propose the following methodology. Intermediate trips and spontaneous identification of vulnerable populations during emergency evacuations are, first, linked with the prevalent paradigm of shared-mobility services (e.g., Uber, DiDi). Secondly, the feasibility of incorporating mobility sharing into emergency-response systems has been discussed by synthesizing attitudes of both the supply and demand sides as well as analyzing factors that influence acceptance of the use of shared-mobility services as an emergency response system. Despite the overall favored results on shared mobility usage, a series of obstacles were identified. Some obstacles would be solved by actionable strategies proposed. However, some obstacles (e.g., panic for the operators and users) still remain to be solved. Next, a quantitative validation has been performed to demonstrate that evacuation efficiency can be improved by incorporating shared mobility into emergency-response procedures, whilst also highlighting the role of shared mobility as an indispensable component of evacuation planning with sound legislation, a complement of current evacuation planning. Figure 1 summarizes the outline of the proposed research.

The remainder of this manuscript has been organized, as follows. Section 2 describes the emergence and development of shared mobility in China and defines the problem of interest. Section 3 introduces the methodology employed in this study, whereas the analysis of results obtained is described in Section 4. Section 5 discusses numerical simulations that were performed in this study. Lastly, findings of the proposed study are summarized in Section 6.

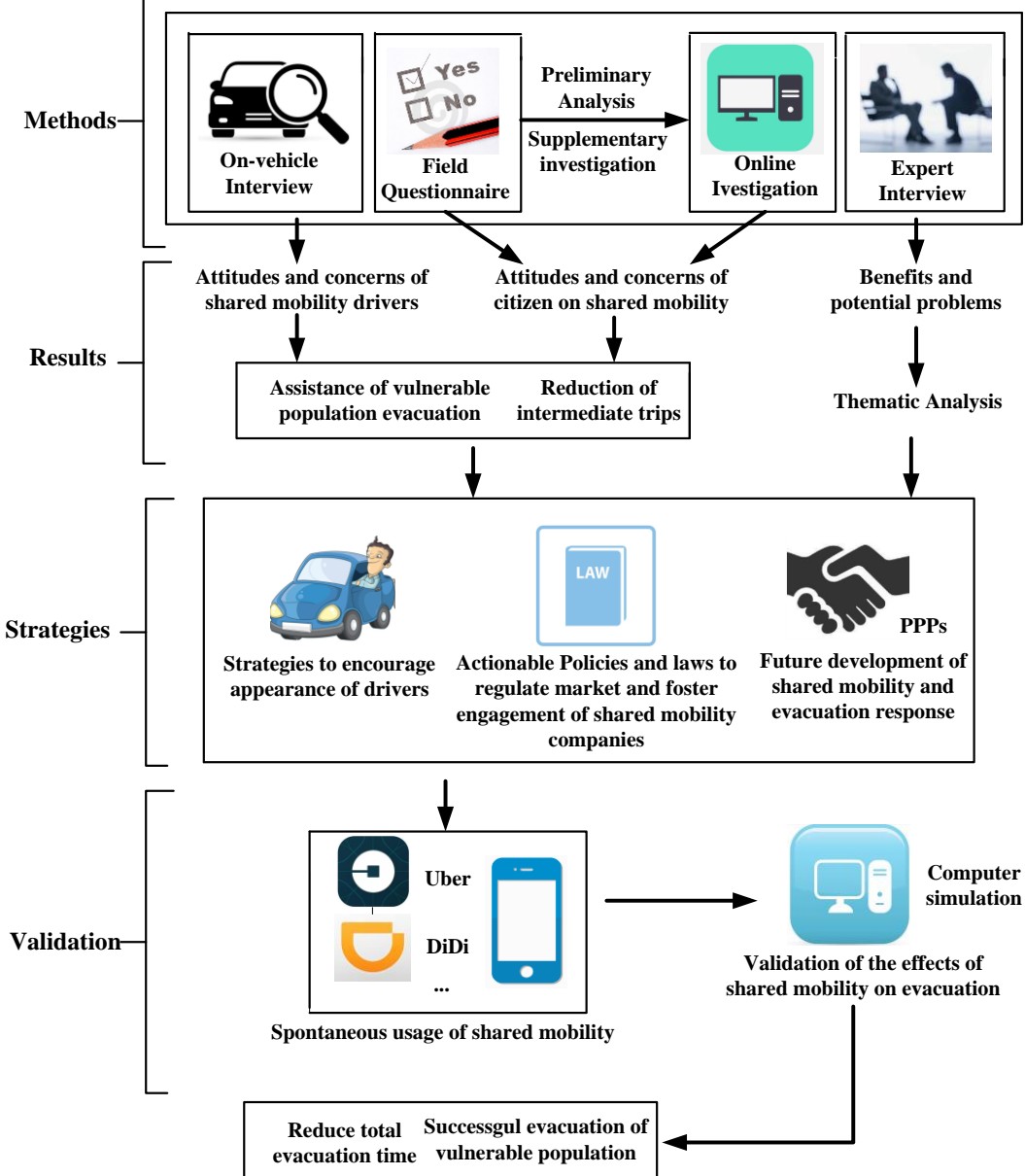

**Figure 1.** Flowchart of proposed research concerning use of shared-mobility services during emergency evacuation.

## 2. Advent of Shared Mobility and Major Concerns Associated Therewith

### 2.1. Emergence and Development of Shared-Mobility Services

With the emergence of the concept of shared economy, the beginning of an era of enhanced intelligence has been witnessed, increased prevalence of which has resulted in the idea of "shared lifestyles" becoming widely accepted among public. Shared mobility has led to significant changes in the mode choice of daily trips that are undertaken by people in China. Especially, the one-to-one service that was provided by shared-mobility operators has changed the pattern of the traditional travel structure. This changes has been brought about by mobile platforms run by businesses, such as DiDi and Uber. Although vehicle-sharing concepts introduced by companies, such as Uber and DiDi, are yet to receive widespread legal acceptance globally, these ideas have become highly popular in China. Appropriate regulation polices and government standards have also been established in this regard in recent years. According to statistics, as of 22 September 2018, 218 out of 297 prefecture-level

cities in China have issued detailed regulations concerning the use of shared-mobility services, since the implementation of "interim measures for shared mobility" issued by the Chinese government on 28 July 2016.

In China, a number of shared-mobility platforms, in addition to DiDi and Uber, have been established in recent years. These include the Shenzhou, Caocao, Shouqi, etc., and the competition between these companies is expected to lead to a more sustainable and healthy socialist market economy. Shared mobility offers convenience and an enhanced commuting experience to passengers, especially in suburban areas with the availability of few vehicles and urban areas during rush hours. Different reward mechanisms have been implemented to promote the prevalence and acceptance of shared-mobility services and stimulate car owners to register as drivers. Considering DiDi as an example, drivers therein can obtain a reward of 45 yuan (approximately 6.7 USD) if they complete 25 orders in a given day. Enhanced rewards are also available for new drivers. Moreover, users can avail themselves of coupons upon use of mobile applications offering discounts on the cost of their next trip. Shared mobility is, therefore, playing an indispensable role in the daily lives of people, and the same can be used to explore potential for more sustainable transportation, given that shared mobility is a flexible resource existing within a region's traffic scheme [35]. The concept of shared mobility has also led to an evolution in the demand pattern structure. As depicted in Figure 2, the shared-mobility industry in China has witnessed rapid development in recent years. Shared mobility accelerates the efficiency of social operations by providing flexible face-to-face and door-to-door services, thereby effectively shortening the daily travel time. Normally, the transportation modes of choice that are considered during emergency evacuations involve private and public vehicles.

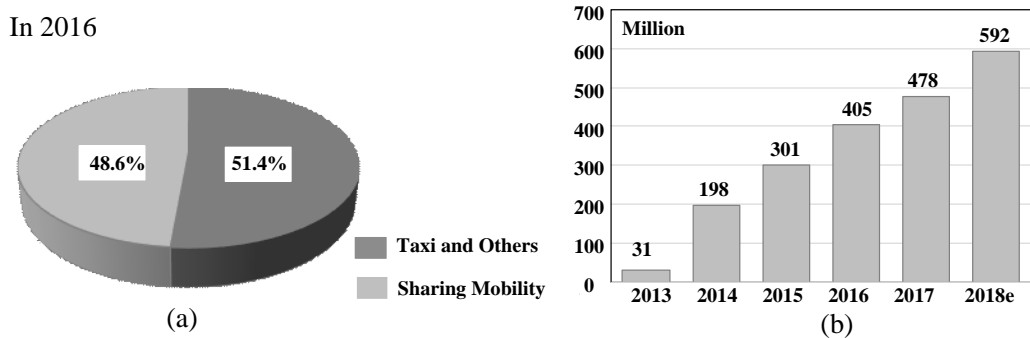

**Figure 2.** Rapid development of shared mobility in China—(**a**) proportion of shared mobility in mobility market (2016); (**b**) increasing scale of shared mobility in recent years.

*2.2. Statistics*

By the end of 2017, the average car ownership in China was less than 0.16 vehicles per person, and the corresponding number in Xi'an was approximately 0.3 vehicles per person. These data indicate the enormous market potential of shared mobility in China in both small and large cities. DiDi alone served over 1.43 billion orders in 2015—one order for every person in China. From a modest 13.6% in 2015, the market share of shared mobility in China rose to 48.6% in 2016, which is almost equal to that of taxis (51.4%). Evidently, shared mobility is slowly becoming the preferred mode of transport for the common public in China. In accordance with expert predictions, the number of shared-mobility users in China is expected to reach 592 million in 2018, and in the near future, a large number of new users is expected to comprise people from small cities and the elderly, who form a significant portion of the vulnerable population. However, despite its great convenience and prevalent usage, there exist certain drawbacks concerning the use of shared mobility. Based on available data, perhaps the biggest problem with shared mobility is the price surge during bad weather. Only 15.2% of drivers maintain their normal price under severe weather conditions. Most prices increase, with nearly 40.2% of the prices rising by a factor of 1.1–2, 33.1% by a factor of 2.1–3, 5.1% by a factor of 3.1–4, and 1.9% by a factor exceeding 4. This surge typically corresponds to the willingness of drivers to operate when

customer demand is high during severe weather conditions. However, most users consider these price surges to be unethical and a form of price gouging, and this can be considered as a significant obstacle during emergency-evacuation scenarios.

*2.3. Main Concerns Relating the Use of Shared Mobility during Emergency Evacuation*

It is important to consider the behavior of shared-mobility operators during emergency scenarios under the coordination of government agencies, such as emergency management agencies (EMAs). In the event of emergency evacuations, shared-mobility operators are expected to make more vehicles available to transport evacuees out of endangered areas and relieve the stress on EMAs with regard to the identification and transportation of vulnerable evacuees. Vulnerable populations that could be assisted by shared-mobility services mainly include transit-dependent groups, such as those of teenagers, low-income individuals, and elderly citizens, who do not live in special-care facilities. As regard vulnerable populations living in special care facilities, such as hospitals, agreements with bus operators seem to be a more feasible option with regard to performing organized evacuation. Highly developed communication technologies and accurate GPS tracking, theoretically, imply that the evacuation of vulnerable groups can be accomplished by shared-mobility services as long as individuals use the relevant mobile application and move onto sidewalks. Hence, the only critical issue that remains is the one concerning an understanding of the level of willingness and the acceptance of responsibility on the part of drivers, passengers, and authorities with regard to the incorporation of shared-mobility services into emergency-response procedures during evacuation. In China, the level of acceptance and availability of shared-mobility services during emergency-evacuation scenarios requires further discussion. Further, the degree of mutual trust between strangers (drivers and passengers) during emergency evacuations remains unknown. To shape a more sustainable future with regard to emergency-evacuation planning and execution, the most challenging aspects relate to the understanding, prediction, and eventually modification of people's mobility behavior.

## 3. Methodology

No field data concerning emergency evacuations in China using the shared mobility are currently available. Consequently, the data required for the analysis proposed in this study could only be obtained by analyzing annual reports of shared-mobility companies, questionnaires, and interviews. Ethical approval for performing this study was obtained from the Institutional Ethics Committee of the Chang'an University, and a questionnaire was prepared to better understand public attitude towards the use of shared mobility during emergency evacuations along with people's reactions under different scenarios. When considering the difficulty that is associated with data acquisition, on-vehicle interviews were conducted to determine the willingness of drivers to provide their services in the event of disasters. Expert interviews were conducted to identify important themes concerning the relationship between emergency response and the use of shared mobility during disasters as well as understanding the benefits and problems that are associated with the incorporation of shared mobility into the scheme of emergency-response procedures, not to mention the government attitude towards the same. All participants and interviewees agreed to the public release of data prior to responding to the questionnaire or interview. All experts that were interviewed in this study agreed to their opinions being used, and the identity of all respondents was maintained anonymous to protect their privacy.

*3.1. Questionnaire Interview*

On-field questionnaire-based interviews on the attitudes of citizens with different socio-economic backgrounds towards evacuation scenarios involving the use of shared-mobility services were conducted between 30 August and 25 September 2017, in the urban area of Xi'an and its vicinity. After a preliminary analysis on sociodemographics and districts, an online questionnaire-based investigation was conducted as a supplementary investigation from 1 October to 15 October. In total, one thousand and five hundred field questionnaires were handed out to citizens in different districts of

Xi'an, and five hundred online questionnaires were emailed to citizens with target attributes by hiring services of a questionnaire company. Of these, 1412 field questionnaires and 479 online questionnaires were collected for further analysis. The online questionnaire was used to make up for deficiencies in information included in field questionnaires (e.g., lack of the number of parent samples, certain levels of annual income, and so on). The response rate of questionnaires was, in general, observed to be relatively high. However, certain questions were asked in a very general manner (for example, age and household income of individuals were not precisely identified, but were rather asked to be within specific ranges). Nevertheless, some responses were incomplete because respondents refused to answer certain questions (e.g., nine respondents were not willing to tell their ages). Geographic locations of participants were almost fairly distributed within each region of the Xi'an city. The percentage and total number of married couples attempting the questionnaire were considered to be sufficient to analyze their stated preference concerning use of shared mobility in evacuation scenarios as a substitute for family pick-ups. The number of participants with no concern regarding family pick-ups was also considered sufficient to analyze their attitudes towards the said use of shared mobility.

Table 1 describes several of the variables (explanatory and decision-making variables) and their descriptive statistics in the questionnaire. The questions shown to each respondent were dependent on the answers given. For example, a respondent who was already happily using shared mobility would not see questions about potential stimuli to encourage the use of shared mobility. In contrast, questions about adverse strategies may be presented to test the level of willingness toward shared mobility.

More specifically, questions that are related to emergency-evacuation scenarios were introduced, as follows. Participants were asked about their planned destinations and the preferred mode of transport—private vehicle, public transit, or family pick-up. Respondents selecting private vehicles as the preferred mode of transport were asked whether they would pick up other family members and make intermediate trips. All of the respondents were asked if they would choose shared mobility as a means of evacuation in the event of an emergency followed by what they would consider an acceptable price, i.e., how much surge pricing would they consider to be acceptable in an emergency scenario or how much decrement in sure pricing would be required for them to make use of the shared-mobility service. Respondents who answered that they would wait for family members to pick them up were asked to choose from various traffic conditions and shared-mobility options. The influence of government policy was also included in the questionnaire under the pretext of "governmental support". The results are analyzed in Section 4.

**Table 1.** Variable definitions and descriptive statistics (partial).

| Variables | Sample Size | Mean |
|---|---|---|
| Personal information of respondents | | |
| Age 18 to 30 (Yes = 1, No = 0) | 1882 | 0.33 |
| Age 31 to 45 (Yes = 1, No = 0) | 1882 | 0.48 |
| Age 45 to 60 (Yes = 1, No = 0) | 1882 | 0.13 |
| Over 60 (Yes = 1, No = 0) | 1882 | 0.07 |
| Gender (Male = 1, Female = 0) | 1891 | 0.58 |
| Education (College and above = 1, Else = 0) | 1870 | 0.48 |
| Marital status (Married = 1, Else = 0) | 1872 | 0.70 |
| Parental status (Parent of child under age of 16 = 1, Else = 0) | 1862 | 0.67 |
| Car ownership (Yes = 1, No = 0) | 1870 | 0.36 |
| Commute mode (Drive = 1, Other = 0) | 1870 | 0.31 |
| Driving experience | 673 | 7.5 years |
| Daily pick-up habit (Spouse or children) (Yes = 1, No = 0) | 580 | 0.73 |
| Daily pick-up habit (Spouse only) (Yes = 1, No = 0) | 580 | 0.12 |
| Daily pick-up habit (Children only) (Yes = 1, No = 0) | 580 | 0.42 |

**Table 1.** *Cont.*

| Variables | Sample Size | Mean |
|---|---|---|
| Household information | | |
| Household car availability (Yes = 1, No = 0) | 1870 | 0.65 |
| Household income less than 100,000 RMB per year (Yes = 1, No = 0) | 1853 | 0.34 |
| Household income of 100,000 to 200,000 RMB per year (Yes = 1, No = 0) | 1853 | 0.49 |
| Household income of 200,000 RMB to 300,000 per year (Yes = 1, No = 0) | 1853 | 0.12 |
| Household income of over 300,000 per year (Yes = 1, No = 0) | 1853 | 0.05 |
| Number of cars in the household | 1870 | 0.77 |
| Choice in emergency scenario | | |
| Evacuation mode (Drive = 1, Other = 0) | 1870 | 0.335 |
| Drive for spouse pick-up (Yes = 1, No = 0) | 532 | 0.79 |
| Drive for children pick-up (Yes = 1, No = 0) | 515 | 0.84 |
| Pick-up distance of spouse | 420 | 7.5 km |
| Travel time for spouse pick-up | 420 | 28.5 min |
| Pick-up distance of children | 432 | 3.2 km |
| Travel time for children pick-up | 432 | 16.6 min |
| Expect pick-up from family members (Yes = 1, No = 0) | 542 | 0.81 |
| Attitude towards shared mobility | | |
| Confidence in shared mobility (Yes = 1, No = 0) | 1891 | 0.71 |
| Experience of shared mobility usage (Yes = 1, No = 0) | 1891 | 0.88 |
| Daily shared mobility usage habit (Yes = 1, No = 0) | 1891 | 0.31 |
| Daily shared mobility commute (Yes = 1, No = 0) | 1891 | 0.08 |

Note: Most families with children at school were single-child families owing to the birth-control policy imposed by the Chinese government over the past century. The two-child policy has only been implemented for the past three years; thus, the questionnaire set contains no detailed information concerning the number of children in a household.

### 3.2. On-Vehicle Driver Interviews

Technically, the relationship between shared-mobility companies and drivers is different from typical employer–employee relationships. Shared-mobility companies provide a platform to registered drivers and earn profit by supplying them with orders. Thus, understanding the attitude of drivers and not shared-mobility companies is more directly linked with the availability of shared-mobility services during an emergency evacuation. The best way to understanding driver attitudes towards and opinions with regard to providing their services during emergency evacuations is to conduct an interview while using the service on the way to and from distributing public questionnaires. In this manner, not only was cost saving realized with regard to arranging individual driver interviews, but also a relaxed and natural atmosphere was realized for conducting interviews, thereby making it easier to understand driver attitudes. Eighty-four on-vehicle interviews were conducted (70 males, 14 females). Some drivers refused to participate in the interview for different reasons, but mostly out of concerns that are related to individual privacy. The interviews lasted 10–40 min, depending on traffic conditions and the travel distance with the average interview time being approximately 22 min. The average driving experience since registration among drivers who completed the interview was approximately 2.5 years (ranging from three months to four years). Besides questions on socio-economic characteristics of drivers, a series of hypothetical emergency scenarios were described to drivers, each has different level of risk in terms of available evacuation time and range of evacuation area. The interviews contained questions related to drivers' willingness and major concerns with respect to providing services during emergency evacuations under different scenarios. Reasons behind the interviewed drivers' registering with shared-mobility companies varied, from earning extra money to making the best use of their free time. As revealed in the interviews conducted in this research, some young single drivers even considered driving as an opportunity to find potential partners. The reason behind the overwhelming gender imbalance of the interview sample could be attributed to the traditional

societal belief that males must bear greater economic responsibility in a family, and that they are more willing to take up driving as an occupation.

### 3.3. Expert Interviews

In addition to the positive attitude of evacuees (car owners and their family members) and drivers towards the role shared mobility can play during evacuations, government opinion is also important to ensure systematic incorporation of shared-mobility services into evacuation-response systems. For a socialist market economy, like China, the market requires macroscopic readjustment and execution of control by the government to result in more robust performance and development. Government intervening can strengthen the administration of registered drivers willing to provide service to facilitate normal operation of shared-mobility platforms during evacuation. Thus, in addition to transportation experts, evacuation experts, natural-disaster policy makers, and socialists, experts from the government and shared-mobility companies were considered when conducting expert interviews. Twenty-five experts accepted the interview invite and provided professional opinions on the concerned topic.

The interviews mainly contained their attitudes and associated reasons on the absorption of shared mobility into emergency-response procedures as well as possible establishment of appropriate public–private partnerships (PPPs) between public agencies and shared-mobility companies. Four members of the research team were responsible for conducting these interviews. Two interviewers (one male, one female) were assigned to each group, with one of them being in charge of the interview and the other responsible for recording the same. All 25 interviews were recorded with due permission of the experts. The average interview time was approximately 20 min and dependent on the availability of the concerned experts. All interviews were completed by 25 November 2017. The interview sample size is justified by interviewing participants until attaining the point of "data saturation", which refers to that point in the data-collection process at which no new additional data can be found to develop aspects of a conceptual category [36]. In this study, Data saturation was ensured by identification of no new themes, findings, concepts, or problems becoming evident in data obtained from three successive interviews after the completion of the first 10 interviews [37]. Hence, the sample size of 25 expert interviews was considered to be sufficient to report important themes with regard to emergency management and the realization of government–shared-mobility partnerships during disasters.

The thematic analysis was proceeded based on the procedures proposed in [38]. Based on their work, six phases should be contained to develop a thematic analysis, which contains data familiarization, initial coding, theme searching, theme reviewing, theme defining and naming, and report producing. First, (1) collecting and familiarizing the data. After this, (2) coding interesting characteristics of the data in a systematic pattern, then (3) collating codes into candidate themes. Further, (4) making a thematic map to illustrate the relationships between codes, between potential themes. Subsequently, (5) generating clear definitions and names for each theme by detail analysis. At last, (6) selecting valuable extract examples and analyze them in relation to research question. The framework for thematic analysis is shown in Figure 3.

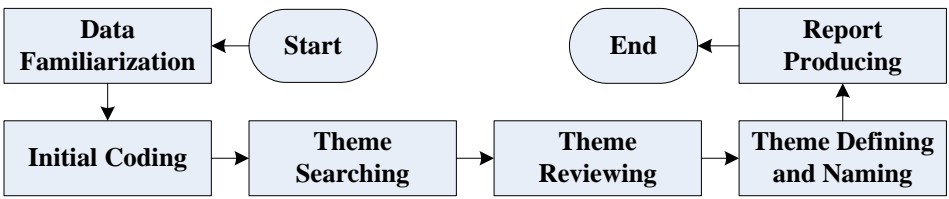

**Figure 3.** Framework for thematic analysis.

As described in Table 2, a relatively high proportion of respondents (64%) felt positive that shared mobility could play an important role in emergency scenarios, thereby supporting the authors' initial idea of integrating the two. However, five experts did not support the proposed idea owing mainly to potential problems that are caused by cooperation or implementation, such as drivers responding to calls, deficiencies in existing PPP policies, concerns regarding mutual trust between strangers, and congestion caused by vehicle reentrance into evacuation zones. Details regarding these concerns and strategies to address the same have been discussed in the next section.

**Table 2.** Overview of expert backgrounds and perspectives.

| Expert Interviews | | | | | |
|---|---|---|---|---|---|
| **Employment** | **Number** | **Domain** | **Number** | **Attitude** | **Number** |
| Universities (Transportation and Sociology) | 9 | Evacuation | 6 | Very positive | 7 |
| Emergency management (Government) | 5 | Policy making | 4 | Mostly positive | 9 |
| Private companies (Shared mobility) | 6 | Disaster response | 5 | Neutral | 4 |
| Public agencies (Transportation) | 5 | Transportation | 6 | Mostly negative | 3 |
| | | Sociology | 4 | Very negative | 2 |
| Benefits of shared mobility usage in emergency response (mentioned) | | | Potential problems of shared mobility (mentioned) | | |
| | **Number** | | | | **Number** |
| Saving budgets for emergency response (service) | 19 | | Government policy (authorities and companies) | | 14 |
| Extra resources for emergency evacuation | 16 | | Social responsibility (companies and drivers) | | 12 |
| Flexible and abundant resource | 14 | | Family concerns (drivers) | | 10 |
| Information sharing | 13 | | Inter-agency cooperation (authorities and companies) | | 8 |
| Communication improvements with evacuees | 11 | | Degree of prevalence in vulnerable population (evacuees) | | 7 |
| Vulnerable population identification | 5 | | Safety of drivers | | 7 |
| Intermediate trip reduction | 3 | | Price surge (companies and drivers) | | 7 |
| | | | Communication (drivers and evacuees) | | 5 |
| | | | More congested situation (drivers) | | 4 |
| | | | Mutual trust (drivers and evacuees) | | 4 |
| | | | Agreements of service during emergency evacuation (companies and drivers) | | 3 |

## 4. Results Analysis

This section analyzes general tendencies of respondents and interviewees, and discusses the major findings that were obtained from interviews and questionnaires.

### 4.1. Major Findings from Questionnaire Response

Responses to the questionnaire reveal that the use of widely spread shared-mobility resources represents a feasible solution for reducing the number of unnecessary intermediate trips during evacuation. As a product of the information age, one-to-one communication between drivers and passengers via mobile platforms demonstrates significant potential for identifying the locations of and evacuating vulnerable groups.

4.1.1. Performance of Shared-Mobility Services as a Substitute for Intermediate Trips

Without considering shared mobility as their preferred mode of transport during initial evacuation, most car owners opined that they would pick up those family members not having access to private vehicles, regardless of the risk (represented by different evacuation deadlines) that is involved. This response can be interpreted as an influence of social attachment [39]. Overall, questionnaire responses revealed that most parents would pick up their children regardless of their daily pick-up habits. Moreover, the pick-up behavior towards spouses was observed to be dependent on the car-ownership status of partners.

Table 3 describes the influence of shared-mobility usage and government encouragement on the reduction of intermediate trips based on the distance of travel and gender difference. Note that the travel distance and government encouragement play an important role in the application of shared mobility during evacuation.

**Table 3.** Reduction in intermediate trips under different scenarios with concerns relating to gender and distance.

| Scenarios | Gender | Object | Number of Family Member Pick-Ups under Different Distance | | | | |
| --- | --- | --- | --- | --- | --- | --- | --- |
| | | | **0~5 km** | **5~10 km** | **10~15 km** | **>15 km** | **Total** |
| Without usage of shared mobility (available evacuation mode: drive, public transit) | Male | Partner | 70 | 145 | 54 | 13 | 282 |
| | | Children | 200 | 65 | 40 | 9 | 314 |
| | Female | Partner | 28 | 91 | 17 | 2 | 138 |
| | | Children | 89 | 25 | 4 | 0 | 118 |
| | Total | Partner | 98 | 236 | 71 | 15 | 420 |
| | | Children | 289 | 90 | 44 | 9 | 432 |
| Shared mobility (available evacuation mode: drive, public transit, and shared mobility) | Male | Partner | 63 (10%) | 99 (31.7%) | 20 (63%) | 5 (61.5%) | 187 (33.7%) |
| | | Children | 177 (11.5%) | 48 (26.2%) | 29 (27.5%) | 5 (44.4%) | 259 (17.5%) |
| | Female | Partner | 15 (46.4%) | 21 (77%) | 8 (53%) | 0 (100%) | 44 (62.7%) |
| | | Children | 72 (19.1%) | 15 (40%) | 2 (50%) | 0 (-) | 89 (24.6%) |
| | Total | Partner | 78 (21.4%) | 120 (49.2%) | 28 (60.5%) | 5 (66.7%) | 231 (45%) |
| | | Children | 249 (13.8%) | 63 (30%) | 31 (29.5%) | 6 (33.3%) | 349 (19.2%) |
| Shared mobility with government encourage (available evacuation mode: drive, public transit, and shared mobility) | Male | Partner | 30 (57.1%) | 34 (76.6%) | 3 (94.4%) | 1 (92.3%) | 68 (75.9%) |
| | | Children | 161 (19.5%) | 43 (33.8%) | 24 (40%) | 4 (55.6%) | 232 (26.1%) |
| | Female | Partner | 8 (71.4%) | 9 (90%) | 1 (94.1%) | 0 (100%) | 18 (87%) |
| | | Children | 47 (47.2%) | 7 (72%) | 1 (75%) | 0 (−) | 55 (53.4%) |
| | Total | Partner | 38 (61.2%) | 43 (81.8%) | 4 (94.3%) | 1 (93.3%) | 86 (79.5%) |
| | | Children | 208 (28%) | 50 (44.6%) | 25 (43.2%) | 4 (55.6%) | 261 (39.6%) |

Post introduction of shared mobility into evacuation procedures, a significant change was observed in spouse-pick-up behavior (a 45% decrease). In contrast, the corresponding change in child-pick-up behavior is less noticeable (19.2% decrease). The observed changes are mainly a result of the difference in trust in the two scenarios and the habit that is associated with availing shared-mobility services, which is a result of their incredible popularization in recent years. Figure 4 illustrates the

degree of intermediate-trip reduction realized under different scenarios based on the travel distance. Profiles that are depicted in Figure 4 were obtained by performing polynomial curve fitting thrice. In general, the degree of reduction in spouse-related pick-up behavior could be positively correlated to the distance traveled to make pick-up trips. The correlation between distance and pick-up behavior could be interpreted as a trade-off between safety and family gathering. Car owners would estimate the time that is required to pick-up their partner based on prevalent traffic conditions. If it would take too long, there existed an unpredictable risk of both partners being present, or worse, stuck within the endangered area. Under such a scenario, therefore, separately evacuating and reuniting in a safe zone seems to be the most plausible solution. An interesting finding is that if the use of shared mobility is encouraged by the government as a policy, the spouse pick-up rate was observed to dramatically decrease by 79.5%, especially for long pick-up distances exceeding 10 km (94% decrease). Therefore, the role of the government is critical with regard to organizing implementable strategies concerning emergency evacuations. Apparently, trends in shared-mobility usage can effectively influence spouse-related family-gathering behavior with results demonstrating that fewer cars in the evacuation network drive long distances to undertake intermediate trips. The reason for only moderate changes in the pick-up behavior for children relates to age anxiety with a considerable number of cases (59%) involving children under the age of 13 years. For cases involving children aged 18 years and above, a trend that is similar to the spouse pick-up scenario can be observed. For children under 13 years, parents would, understandably, wish to pick them up unless their safety can be ensured by means of implementing convincing strategies on the part of the government.

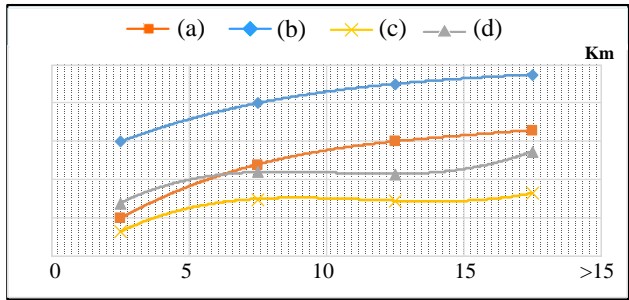

**Figure 4.** People's willingness to use shared mobility with respect to distance under different scenarios—(a) spouse pickup; (b) policy-based spouse pickup; (c) children pickup; and, (d) policy-based children pickup.

In addition, the influence of trip distance on children pick-up behavior demonstrated differences on male and female responses (as depicted in Figure 5a,b). As regard female responses, the degree of change in children-pick-up behavior can be positively correlated to the trip distance. However, as regard male respondents, the observed children-pick-up behavior was not as sensitive as that of females, and the general level of change was less when compared to female responses. In terms of spouse-pick-up behavior, the observed variation trend was similar among both male and female respondents.

Reference to Table 1 demonstrates that the average pick-up time for children or spouse is much longer when compared to the average waiting time for shared-mobility vehicles, which is approximately 5 min. Moreover, according to statistical data, under normal circumstances, the average travel distance (less than 1.5 km) between shared-mobility vehicles and car-less evacuees is much shorter as compared to the that of intermediate trips made by family members; thus, the total trip distance and travel time of vehicles during an evacuation can be effectively reduced via the use of shared-mobility services, thereby further enhancing evacuation efficiency and reducing network clearance times. Validation of this hypothesis has been introduced in Section 5.

In conclusion, the use of shared-mobility services with government encouragement can effectively reduce the number of intermediate trips concerning grown-up family members, especially in cases involving long trip distances. Evacuees with longer trip distance bear are subject to greater uncertainties during evacuation. Therefore, a reduction in the number of such long trips is an urgent requirement from the viewpoints of evacuee safety as well as EMA management. Note that the ubiquity of smartphones among teenagers is an important prerequisite for validating the use of shared mobility.

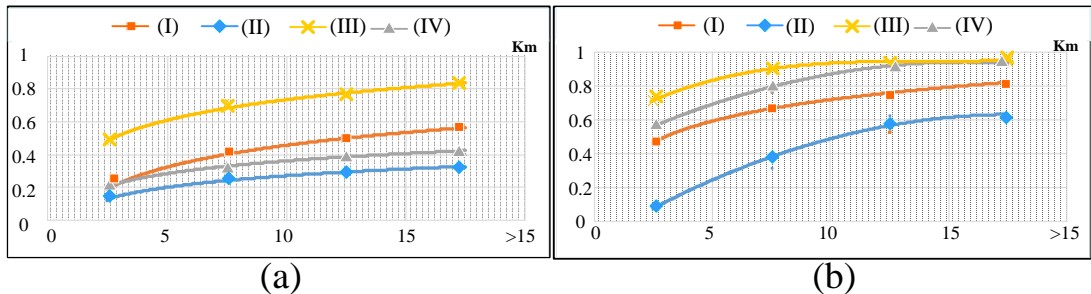

**Figure 5.** Degree of change in (**a**) children and (**b**) spouse pick-up behavior—curve (I) female response (without government encouragement); (II) male response (without government encouragement); (III) female response (with government encouragement); (IV) male response (with government encouragement).

### 4.1.2. Single Vulnerable Evacuees

As already mentioned, vulnerable populations that would reap the most benefits out of utilization of shared-mobility services would most benefit are mainly groups of transit-dependent people, teenagers, low-income individuals, or elderly people not living in special care facilities. Based on data acquired via the questionnaire, the tendency to avail of shared-mobility services among these vulnerable populations is mainly influenced by the price of and time available for evacuation (i.e., level of urgency). Note that, for a free and low-price market, the use of shared mobility is popular among younger generations (aged 28 years or less) of non-native citizens, i.e., those who live alone and have no family members residing in the city (e.g., college students, graduates, and alien workers). The average age of this group (24 years) is much less when compared to that of the remaining population (40 years). Based on expert opinion and a series of telephone inquiries, major reasons behind the high acceptance of shared-mobility services among this group include—(a) relatively less social attachment; (b) low rate of car ownership; (c) low income; and, (d) open-minded approach to new techniques and lifestyles. Moreover, the acceptance rate in surge-pricing scenarios depends on whether these non-native citizens are staying with friends or classmates. This phenomenon can be interpreted as an apportionment of expenses. For middle-aged commuters, the choice is insensitive to the price of shared-mobility services. The acceptance of shared mobility among the elderly is relatively low, owing to rather little smartphone usage; however, this age group demonstrated willingness to use the apps if necessary, although some of them are not familiar with making e-payments.

### 4.1.3. Modal Split

The majority (93%) of the evacuees owning cars chose to drive out of the evacuation zone. Only a small portion of car owners chose not to drive owing to their rather little driving experience; this could be inferred from their answers to other questions regarding whether they have family members in the city or have chosen to wait for pick-ups (female respondents). Besides experienced drivers, family gathering and pick-up behavior were also evident in the responses of less-experienced drivers. All 11 respondents with little driving experience but having family members that are unable to drive chose to drive so as to pick them up. Among the respondents (626) choosing to drive to evacuate, 85% (532) were married and lived with their family, whereas 515 of them had children and lived

together. Of the 532 respondents, 420 (79%) would pick-up their spouse and 432 (83.9%) of the 515 respondents would pick-up their children before driving to the final destination. This percentage was not remarkably high because certain households owned two cars; thus, the child pick-up mission was shared by parents. Interestingly, an overwhelming majority (92%) of spouses assigned to that mission were male. If we consider a household as a unit, the number of family gatherings before heading to the final destination nearly equaled 100% for households with car owners.

When compared to public transit, most car-less evacuees (81%) from households owning at least one car expected to be picked-up by their family members, but once shared mobility became an alternative, a majority (83%) of these car-less evacuees choose to avail of shared-mobility services, owing to the shorter waiting time and evacuation duration involved as well as out of safety concerns for their partners. This proposition is tenable on the premise of mutual agreement or tacit understanding between car owners and car-less family members concerning the utilization of shared mobility. If conflicts exist between car owners and car-less family members in the use of shared mobility, it will be difficult for shared-mobility services to make a significant difference with regard to reducing intermediate trips during emergency evacuation.

Besides attitudes of car owners and their car-less family members, choices that are made by other population are also important to validate the use of shared mobility during evacuation. Most car-less evacuees (95%) habituated with public-transit commute and had no household-owned private vehicles choose public transit as their preferred mode of evacuation mode as long as the service remained in operation; and, their choice was insensitive to the discounted price of use of shared mobility (a decrease from 95% in the normal-price scenario to 90% in 50%-discount scenario). The relatively low acceptance rate of shared mobility among this group of people was mainly caused by reasons pertaining to—(a) personal habits (the group was habituated to public-transit commute and demonstrated greater trust in public transit); (b) psychological reassurance (people tended to stay with the majority to "feel" safe as emergency situations result in higher mental stress; in other words, they wish to stay with the crowd); (c) availability anxiety (as compared to public transit, the certainty of shared-mobility pick-ups cannot be assured); and, (d) price (compared to fare-free public-transit service during evacuation, the high price of the shared-mobility services is not acceptable). This finding was considered positive under emergency-evacuation operations, since it would prevent further congestion being caused by high demand levels of shared mobility among public-transit commuters; i.e., shared mobility would only be required to provide services to evacuees less likely to choose public transit during evacuations. There, therefore, exists a high possibility of the main users of shared-mobility services belonging to the group of car-less evacuees awaiting pick-up by family members.

Based on the analysis presented in this section, a consensus was achieved with regard to car-owners and their grownup family members using shared mobility as an alternative mode to substitute family-member pick-ups during evacuation scenarios. Moreover, there existed no concerns regarding competition among commuters (evacuees expecting pick-ups by family members and public-transit commuters) to avail of shared-mobility services. These findings confirm the feasibility of incorporating shared-mobility services into emergency-response procedures from the demand side. The feasibility from the supply side has been discussed in the following subsections.

*4.2. Understanding Willingness and Concerns of Shared-Mobility Drivers*

To valid the use of shared mobility during evacuation, the basic requirement is the willingness of service providers to participate in the evacuation. It is, therefore, worth understanding the willingness and concerns of shared-mobility drivers to provide service during emergency situations. Moreover, this study also aims to understand what attributes make a driver trustworthy to provide service during evacuation, and under what conditions would a driver be willing to provide service during evacuation.

In many cases, shared-mobility drivers are evacuees themselves, and their family members might also be in the area affected by the disaster. Therefore, the willingness of drivers to provide service during emergency evacuation must be divided into three scenarios—(1) the driver and his/her family

members are not in the area to be evacuated; (2) the driver's family members are in the endangered area; and, (3) only the driver is currently in the evacuation area. In the first scenario, the drivers need to drive into the endangered area. In the second scenario, drivers could be considered as evacuees with access to automobiles. However, after safely evacuating family members and provided that there still exists time, the situation of drivers in scenario two becomes similar to that in the first scenario. The third scenario implies that drivers can pick-up evacuees on their way out of the evacuation zone, which seems like the most convenient situation if they choose to provide a service because it starts at the latter part of the first scenario, albeit with more buffer time prior to the impact of the disaster.

The application of a thematic analysis framework should be on premise of the selection of the most lively and attractive extract examples of individual responses, as shown in [40]. As the respondents speak Chinese, therefore, the extract examples are translated to English by the authors to make them understandable by readers. The data are anonymized to be in line with typical qualitative research ethics norms, that is, the surnames in the extract examples of individual responses are not real. This statement is also applied in the part of analysis of expert interviews.

### 4.2.1. General Attitudes under Different Scenarios

Based on responses obtained from drivers during interviews, the most adverse situation for drivers to provide service corresponds to the second scenario. Most drivers demonstrated an unwillingness or reluctance to provide a service during the second scenario. They cared more about the safety of their own family members, especially male drivers with children under the age of 16 years, because being a driver of a shared-mobility company is not their formal job. Once the safety of their family members is assured, most male drivers (52 out of 59) are willing to provide service to other carless evacuees if there is enough time remaining to help them (i.e., they would pick-up passengers nearest to them). They believe that their spouses and children can take care of each other in a safe place, and their reason for providing service to car-less evacuees is either for availing of a financial bonus or self-actualization.

*"It is not a problem to provide service during emergency evacuation, if my family members are safe, and . . . hum, we will be safe, right?" Mr. He, aged 42, married.*

*"I think I will save my family members first, uh-huh . . . but I will consider to do so after my family are safe, and . . . can I get pay for this?" Mr. Zhao, aged 34, married.*

However, most female drivers (12 out of 14) would prefer to stay with their family members in a safe place rather than participating in the evacuation service.

*"It is less likely for me to drive into the dangerous area, this is the job for firemen . . . That is not to say I don't want help them, but . . . I am a woman, I want my family safe . . . It is just that I don't think I can finish the task!" Mrs. Wang, aged 41, married.*

In the third scenario, it is encouraging to find that most drivers regard their service as a responsibility and pay less attention to profit. However, price surges are an effective method of stimulating drivers to provide service in the first scenario. Without this stimulation, drivers could tend to avoid risk by only providing service to passengers that are located outside endangered areas. Moreover, in the event of all orders outside the endangered area being canceled by shared-mobility companies, drivers would also provide service to car-less evacuees within the endangered zone.

*"The company should consider our safety if they choose to do this. Providing service to evacuees in danger is more than a job; it's heroic" Mr. Gu, aged 46, married.*

The effect of the shared-mobility platform is also evident in the behavior of drivers once the safety of their family members has been ensured.

### 4.2.2. Important Role of Single Young Male Drivers

Among the 84 drivers accepting to be interviewed, 11 were single male drivers. Single female drivers accounted for only a very small portion of the total numbers of drivers registered with shared-mobility platforms; thus, data that were obtained from only single male drivers are discussed. The willingness of single, young male drivers to provide service during emergency evacuations is impressive. In the third scenario, 10 of the 11 single, young male drivers demonstrated willingness to provide service to people in need, considering this a journey of adventure for themselves and a means to realize self-actualization. The only driver unwilling to pick up passengers during evacuation cited the fear of disaster. Even when faced with the first scenario, 8 of the 11 drivers agreed to drive into the endangered zone and rescue car-less evacuees. These results suggest that the family-gathering behavior is mainly restricted to those with partners and children.

> *"Actually, I don't think it is a problem for me, I believe I can keep myself from danger, BTW, it sounds really cool if I can save lives like a fireman (laughter)!" Mr. Zhang, aged 25, single.*

> *"If I can save others' lives, why should I hesitate? It is more than an experience of excitement, but a self-actualization" Mr. Zhao, aged 27, single.*

Results of driver interviews effectively support the common and often predominant expression of mutual aid indicated in literature [39]. In conclusion, male drivers are more willing to provide services during emergency evacuations after the safety of their family members is assured. Encouragingly, male drivers accounted for a majority of shared-mobility drivers. A possible means to validate the use of shared mobility during emergency response involves encouraging cooperation between single, young male drivers and EMAs.

### 4.3. Benefits of Shared-Mobility Usage and Solutions to Potential Problems

After the analysis of questionnaire and on-vehicle interviews, the results of expert interviews are discussed in this section. The data are anonymized to be in line with typical qualitative research ethics norms, as claimed before. Despite the progress made and lessons learnt, experts agreed on the urgent need for improvements in emergency management practices. As an emerging lifestyle, shared mobility could possibly play an important role during emergency evacuations by leveraging the organized spare capacity of vehicles as evacuation resources. The extent to which this role can be fulfilled depends on many factors, including the popularity of shared mobility in certain urban areas, characteristics of the emergency, public predisposition, willingness of drivers, amount of vehicles available, and shared-mobility characteristics. To turn this potential into reality, some practical problems still need to be considered and addressed. Involving shared-mobility companies into emergency response would involve PPPs between public agencies and shared-mobility platforms, which also provide future development prospects to shared-mobility companies. Since the experts that were interviewed in this study belonged to from different domains, the topic could be treated from different perspectives.

### 4.3.1. Benefits

Application of shared mobility during emergency evacuations offers multiple benefits for public agencies, shared-mobility companies, drivers, and evacuees. From the perspective of public agencies, the benefit of leveraging shared-mobility services as emergency response is fivefold. First, shared mobility is an extra resource, isolated from current transportation resources used in evacuation planning. This can be regarded as additional vehicle capacity for evacuation, especially for vulnerable populations (e.g., car-less people and those dependent on family pick-ups). Secondly, it is more economical than storing extra resources and matching them with needs, because their daily operation is organized by private-sector stakeholders. Next, the characteristics of flexibility and abundance that are possessed by shared mobility can meet the needs of most potential passengers, so long as

they possess the required mobile applications and can move to the roadside. The participation of shared economy during emergency evacuation can relieve the stress on EMAs in terms of resource distribution and the evacuation of vulnerable populations, especially people who are not familiar with the surroundings and those with health problems. The door-to-door service that is provided by shared-mobility companies is more useful in emergency-evacuation scenarios when compared to normal days. Additionally, the one-to-one communication between evacuees and drivers is an innovation with regard to evacuation scenarios in terms of realizing the accurate location of vulnerable individuals and understanding their special needs. Moreover, shared-mobility vehicles could be treated as small "mobile relief stations" if they were equipped with medicines, food, water, wheelchairs, etc. Although such vehicles are not able to meet special needs of certain segments of the population, drivers can report the situation to companies or EMAs agencies under the PPP background. The new pattern of information sharing can surely enhance evacuation response efficiency. Lastly, the number of intermediate trips could be reduced via incorporation of shared mobility, which can be interpreted as less pick-up behavior within an evacuation network. The total trip distance during evacuation can also be reduced.

*"Before the prevalence of shared mobility, we have sought for feasible ways to identify the location of vulnerable population, but there is no applicable strategy can overcome this problem, but now, I can see a promising future" Mr. Liu, Disaster response.*

*"If the shared mobility can be a part of evacuation response system, the capacity of transportation will be improved, because more vehicle are available to pick up people in need instead of driving alone" Mrs Sun, Transportation.*

*"For those evacuees with health issues, if a part of the vehicle can be equipped with medicines, and even other facilities, that would be perfect!" Mr. Ma, Evacuation.*

As regarding shared-mobility companies, the benefits include positive coverage to the public and potential to attract more customers to help improve their competitive force in the shared-mobility market in cooperation with government agencies. PPPs would help to secure sustainable and healthy development of such companies in future.

*"Just imagine that a company play an important role in the city evacuation, it will be a big news, I think the shared mobility should establish PPPs with government, because this would be a win-win situation for both ends" Mr. Liu, Policy-making.*

With regard to evacuees, shared mobility provides an additional means to help them escape from endangered areas. When compared to public transit, evacuees can save more time and have an enhanced evacuation experience. The amount of intermediate trips replaced by shared mobility can effectively help in reducing the evacuation trip distance of car-owners, thereby realizing greater safety assurance.

*"For evacuees, a more option in emergencies means a higher opportunity to be safe!" Mrs. Li, Sociology.*

From the perspective of drivers, providing this service corresponds to an opportunity to earn extra income and realize a feeling of self-actualization. This is also considered a supreme glory by Chinese citizens.

4.3.2. Potential Problems and Solutions

Although multiple benefits can be realized via incorporation of shared-mobility services into the evacuation response system, multiple problems still exist. A set of implementable policy recommendations has been developed to solve potential problems.

*Policy-related concerns and solutions*

The first problem concerns the universality of shared-mobility usage in different areas, which can be determined based on regional characteristics. Although shared mobility is popular in big cities, the amount of available resources and the traditional outlook of the elderly population in small cities is different from those in big cities, and the usage of shared mobility is still not as prevalent in the Chinese countryside. Regional differences that exist are also noticeable.

> *"Although it might be a plausible solution to solve a series problems, but if the size of this industry is small in some area, it can only make limit effects on evacuation" Mr. Wang, Policy-making.*

Therefore, the use of shared-mobility services as emergency response must be based on the characteristics of the city concerned. Therefore, the implementation of this strategy should be decided by the local authorities instead of the central government. Additionally, relevant studies need to be performed in small Chinese cities.

Emergency incidents could be categorized as short-notice evacuation (e.g., evacuation caused by hurricane Katrina) and no-notice evacuation (e.g., evacuation caused by the "911" incident), depending on whether advance notice is available. The modal choice mechanism is different for the two types of evacuations. Normally, the available evacuation time in no-notice scenarios is less when compared that in short-notice scenarios. The hypothetical evacuation considered in this study corresponds to no-notice evacuation, aiming to save evacuation time and improve evacuation efficiency of the endangered area. Thus, the stated preference results concerning mode choice and shared-mobility availability have also been adapted to the no-notice scenario. However, the availability of shared-mobility platforms during evacuation must be ensured. This calls for an agreement between the government and shared-mobility companies to facilitate normal operation of the platform. The best solution to this problem lies in establishing PPPs between the government and shared-mobility stakeholders.

Additionally, a more important aspect involves ensuring that drivers show up in an emergency situation. This problem is similar to those that are faced during transit planning in the event of an evacuation, as noted by a special report of the Transportation Research Board [41]. For most drivers that are registered with shared-mobility companies, driving is just a part-time job to earn extra money. Whether or not they wish to undertake the social responsibility of providing service to vulnerable populations during emergencies requires further discussion.

> *"How to make sure the driver will show up during emergencies, you know, it is not a formal job, not like the emergency personnel . . . will they be willing to do this job?" Mrs. Zhao, Sociology.*

EMAs must include shared mobility within their planning process and ensure that drivers are available to provide services to vulnerable populations. This calls for agreements between companies and drivers to ensure their timely appearance in the event of an emergency. From the management aspect, prepopulation of a list of registered vehicles from shared-mobility companies is required to support emergency evacuations or make them available on-demand basis. This is similar to volunteer evacuation corps proposed in [42], but with payments and benefits that are meant for their future profits, as the implementation of strategies to encourage the appearance of drivers under emergency situations. To make this available, a questionnaire designed based on data acquired by on-vehicle interviews to be filled in by car-owners when they register with shared-mobility platforms. For drivers meeting the needs of emergency response, an optional agreement with shared-mobility platforms and government is to be provided to them. Based on analysis results that were obtained from on-vehicle interviews, the target group could correspond to single, young male drivers. This group of drivers possessed the least uncertainties and social attachment during emergency situations, thereby making them trustworthy for evacuees and EMAs. Vehicles in this list should be pre-equipped and fitted with resources, such as the first-aid kid, wireless radio, transportation of disabled population, child car seat, stretcher, and so on, to support emergency evacuation. Other vehicles that were owned by

shared-mobility companies participating in the evacuation could provide service to evacuees with no special needs other than transportation.

In detail, the following actionable strategies could be employed to encourage the appearance of drivers.

(a)     Agreements must be signed and reward mechanisms implemented to stimulate drivers to show up during emergency evacuations. For example, 50 Yuan (approximately 7.5 USD) could be offered to drivers for every order to evacuate people from endangered areas, although an accurate figure requires more discussion concerning the trade-offs to be made between willingness and recklessness.

(b)     An honorary title in the mobile application must be awarded to drivers making themselves available during emergencies to improve the quality of future orders (e.g., shorter pick-up distances and orders with better traffic conditions). The honorary title confirmed upon drivers must be made visible to mobile-app users.

(c)     Widespread education of shared-mobility companies to increase awareness regarding social responsibility among drivers.

If drivers to show up, their safety becomes another concern. Protecting drivers from danger is an urgent problem.

*"(Continued) . . . and even though they are willing to provide service, how can we keep them safe?"*
*Mrs. Zhao, Sociology.*

Drivers are not emergency-response professionals; hence, specific training would be required to prevent them from landing into dangerous situations in sensitive areas, thereby reducing their liability and risk. To avoid creating unnecessary congestion, shared-mobility companies should arrange for the closest drivers to pick up evacuees. Fortunately, this function is already being automatically realized through apps. For registered drivers that are willing to provide service, an online curriculum on emergency response and right actions in different scenarios must be issues by EMAs.

Another problem that is associated with the use of shared mobility during emergency evacuations is the price-surge phenomenon, which tends to impede the prevalence of shared-mobility services, especially among low-income evacuees (the group of most concern, as many of them have no access to automobiles).

*"The problem of price gouging during adverse weathers would limit the role of shared mobility, actually, it is a way to encourage drivers to provide service, however, is it reasonable during emergencies?"*
*Mr. Lei, Policy-making.*

Prices must be regulated by the government to prevent price gouging during emergency evacuations. To exercise control over price surges during an emergency, (a) the extra expense should be shared by companies and the government, and the degree of price surge should be regulated by laws to avoid price-gouging; and, (b) drivers should receive an extra bonus because they are acting as proxies for EMAs. This is also an alternative way to encourage the willingness of drivers.

To build mutual trust between evacuees and shared-mobility drivers, the government must issue relevant documents, distribute them among employers, and let them convey contents therein to employees, thereby fostering public trust. In accordance with questionnaire data, shared-mobility services can attract greater social trust upon establishment of PPPs, reflected by substantial increase in preference for shared-mobility services when compared to waiting for family-member pick-ups. Moreover, the government can implement preferential policies for shared-mobility companies to foster their willingness to engage with the government in terms of disaster policies and further realize sustainable management and development. Most citizens believe in the effectiveness of government policies; hence, PPPs are an effective method of guaranteeing the trust of citizens in the reliability of shared mobility for providing evacuation assistance.

*Technical concerns and solutions*

The above constitute major policy-related concerns and corresponding solutions; technical concerns, on the other hand, which are related to the topic herein mainly concern congestion anxiety, appropriate road assignment, and vulnerable populations without access to shared-mobility services.

A major concern expressed by transportation experts is that when evacuation happens, many Uber or Didi cars enter into the evacuation area, thereby causing severe traffic congestions that impact evacuation efficiency. This concern is meaningful in real-life evacuation scenarios. After the occurrence of a disaster, shared-mobility vehicles are naturally expected to enter endangered areas. However, when compared to the amount of partners and parents outside that drive into endangered zones to pick-up family members, the corresponding number of shared-mobility vehicles is rather small. Shared mobility can provide a minimum of four seats to evacuees with their carpooling service as compared to only one or two seats in the family-member pick-up situation. Moreover, in accordance with results of numerical experiments that are discussed in Section 5, trips for family-member pick-ups from non-endangered areas increase rapidly after the commencement of evacuation, whereas the corresponding load on shared-mobility vehicles from non-endangered areas is relative steady. Therefore, the role that shared mobility plays as a substitute for intermediate trips is not expected to affect evacuation efficiency.

It is noteworthy that the service provided by shared-mobility vehicles is different from that of official vehicles dispatched by EMA. The former is more like an agreement between evacuees and shared-mobility companies. Therefore, the rights of shared-mobility vehicles during evacuation must not be superior when compared to those of normal vehicles. Hence, shared-mobility vehicles must also follow special transportation rules, such as road closure, diversion, and prioritization during evacuation scenarios, as compared to official vehicles that are dispatched by EMA to avoid conflicts in traffic management.

Among multiple benefits offered by the use of shared mobility lies the ease of identification of vulnerable population. However, this benefit can only be realized by evacuees with access to shared-mobility mobile applications. Vulnerable populations without access to such apps include elders that do not own smartphone and foreigners. A solution to this deficiency lies in the realization of volunteer-supported and community-based neighborhood assistance or reference to the Citizen-assisted Evacuation Plan (CAEP) for the city of New Orleans [12]. Foreign evacuees can use the international version of apps and directly contact the customer service in case they encounter communication problems relating to inaccurate GPS positioning or other special needs. Subsequently, the nearest qualified vehicle can be assigned to provide the required service. Under situations of inaccurate GPS positioning, the user can directly contact drivers via phone calls and/or messages. Another obstacle is the panic behavior among public and drivers, since the road condition during emergencies are more likely to be chaotic at the beginning of evacuation. Training activities contribute to improve evacuation planning by means of an increase of capability, and they are also important factors in risk reduction in terms of exposure [3,43]. Therefore, to obtain calmness among ordinary individuals during emergencies, a series of special training should be offered to the drivers and public, however, this might not be easy to realize in the short term. It needs wide cooperation among governmental agencies and organizations on different levels. Also, it needs to be verified by the work of [44].

## 5. Numerical Experiments

### 5.1. Objective of Numerical Experiments

Numerical simulations were performed in this study to validate the potential of shared-mobility services to reduce network-clearance times and the total trip distance during evacuations.

*5.2. Preparation and Description*

A hypothetical, no-notice, daytime evacuation scenario was considered requiring the evacuation of the entire population within the affected area of the Xi'an city (Figure 6). Location of the incident was considered in the middle of the city with radius of the affected area being 5 km. The incident was assumed to occur during work hours, i.e., between 0900 h and 1100 h and between 1500 h and 1700 h, to incorporate intermediate trips into evacuation. No shelter-in-place strategy or shadow-evacuation scenarios were considered. Unidentified vulnerable populations have also not been considered. The evacuation scenario was classified into the following four situations that are based on the pick-up behavior and availability of shared-mobility services.

Baseline case—without considering pick-up behavior or use of shared mobility, everyone would choose the most efficient way to evacuate, except when waiting for pick-ups. In this case, schools were assumed to be able to evacuate all children using school buses or other modes of transportation, and adults without cars were to evacuate via the most efficient public-transit mode.

Practical case—without application of shared mobility; that is, all pick-ups would be accomplished by family members. The pick-up rate was derived from questionnaire data, and data relating trip distribution were collected from an output file generated by Dynasmart-P when simulating the disaster scenario.

General shared-mobility case—with application of shared mobility, model data were acquired by analyzing questionnaire and on-vehicle interview data. Some drivers were considered to not be available at the beginning of the evacuation because their family members were also located within the danger area.

Ideal shared-mobility case—family member pick-ups were accomplished by shared-mobility drivers and car owners under system optimization (SO) resource allocation deployment.

To analyze citizen behavior during evacuation, the first step involves locating evacuees within the endangered area according to their work and school locations and matching the number of vehicles available to household members to determine the availability of cars among evacuees. Since actual data concerning the work or home addresses of individuals are difficult to obtain, evacuee distribution within the endangered zone can be derived from responses to the questionnaire. First, respondents were considered to be randomly distributed within the endangered region (radius equal to 6.5 km), and their transportation-mode choice was based on answers that were provided to the questionnaire. Secondly, for married samples, the locations of their partners were also considered to be randomly distributed within a range of distances obtained from answers to the questionnaire (as listed in Table 1), the location of children are assigned to the actual locations of schools within the network at distances within the range obtained from answers to the questionnaire. Subsequently, the data size was expanded to approximate the population of the evacuation area during regular hours. Distribution data concerning shared-mobility vehicles at 1000 h on a regular work day were acquired from Didi and Uber. The vehicles were considered to be present at or near locations specified in data provided by shared-mobility platforms. For the no-notice, daytime evacuation scenario, all evacuations were assumed to require simultaneous on-road loadings. In the baseline and practical cases, shared-mobility drivers within the evacuation area were considered evacuees. Trip chains were considered in the practical case, and pick-up behaviors were acquired based on answers that were provided by respondents to the questionnaire. For evacuees that would stop more than once during evacuation (i.e., during both partner and child pick-ups), the stops were sequenced based on the "nearest-first" principle. In the general shared-mobility case, the pick-up behavior was assigned based on respondent answers to the questionnaire after government encouragement in support of shared-mobility usage. Trip chains of shared mobility were based on the principle of the shortest trip distance. In the ideal shared-mobility case, whether evacuees were picked up by shared-mobility or family members was based on the trip distance instead questionnaire answers. Finally, a dynamic traffic-assignment method was employed to assign time-dependent evacuation chains to the network.

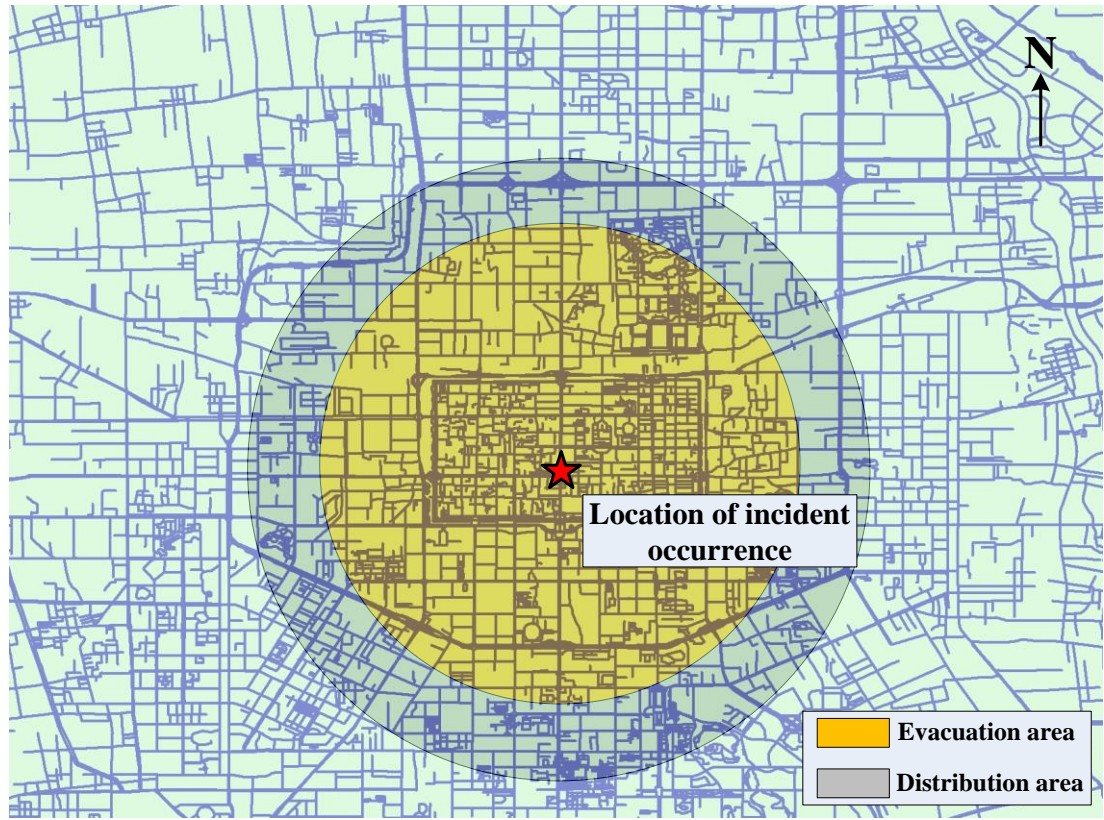

**Figure 6.** Road network and corresponding range of evacuation area considered during numerical simulation.

*5.3. Simulation Results*

Table 4 presents the number of evacuees and total trip distance required for evacuation under the four scenarios considered. An estimated 505,314 people were required to be evacuated in the baseline case without considering intermediate trips. In the practical case, this number increased to 576,352—an increase of 14% when compared to baseline. The associated trip distance increased by 69.6%, owing to the occurrence of intermediate trips. Note that the percentages and numbers mentioned would vary depending on the range of influence and location of incident. Post introduction of shared mobility as an available choice, the total trip distance was observed to have reduced by 21% from 2,821,307 km to 2,230,512 km. The number of evacuees reduced by 8.6%, and the mode selection of driving alone without intermediate trips increased by 53.5%. These results indicate that far fewer car owners drove into the endangered area to pick up family members, and a considerable number of pick-up trips were substituted by shared mobility.

**Table 4.** Number of evacuees and trip distance under different scenarios.

| Scenarios | Mode of Evacuation | Number of Evacuees | Trip Distance (km) |
|---|---|---|---|
| Baseline | Drive alone | 191,692 | 613,414 |
| | Public transit | 254,189 | 889,661 |
| | School bus | 59,433 | 160,469 |
| | Total | 505,314 | 1,663,545 |
| Practical | Drive alone | 101,342 | 324,294 |
| | Pick up and be picked up | 261,357 | 175,1092 |
| | Public transit | 211,322 | 739,627 |
| | School bus | 2331 | 6293 |
| | Total | 576,352 | 2,821,307 |

**Table 4.** *Cont.*

| Scenarios | Mode of Evacuation | Number of Evacuees | Trip Distance (km) |
|---|---|---|---|
| | Drive alone | 155,564 | 497,804 |
| | Public transit | 131,243 | 879,328 |
| General shared mobility | Pick up and be picked up | 199,374 | 697,809 |
| | School bus | 2331 | 6293 |
| | Shared mobility | 38,276 | 149,276 |
| | Total | 526,788 | 2,230,512 |
| | Drive alone | 180,226 | 576,723 |
| | Pick up and be picked up | 104,742 | 377,071 |
| Ideal shared mobility | Public transit | 199,233 | 697,315 |
| | School bus | 2311 | 6239 |
| | Shared mobility | 63,522 | 228,679 |
| | Total | 550,034 | 1,886,029 |

Figure 7 depicts the evacuation performance under different scenarios in terms of the successfully evacuated population with respect to time. As can be seen, the output of traditional evacuation models that do not consider intermediate trips is overly optimistic, even in the ideal shared-mobility case, thereby supporting results that were reported in an extant study performed by [17].

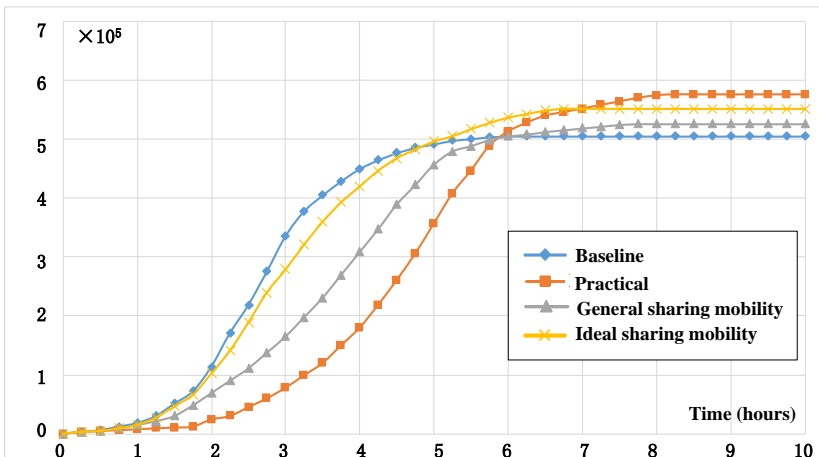

**Figure 7.** Cumulative trends concerning successfully evacuated population against time.

The reason behind the variance in the maximum number of evacuees under the different evacuation scenarios can be explained, as follows. Family members with cars or shared-mobility drivers enter the endangered area and become evacuees themselves whilst undertaking intermediate trips. Nearly all intermediate trips occur at the beginning of an evacuation, which in turn, explains why the successfully evacuated population in practical cases is much lower when compared to the other three cases. For the baseline case, evacuation was completed in approximately 6 h after occurrence of the incident. In the practical case, however, when considering intermediate trips without the use of shared mobility, only 89% of the population had been evacuated up to this point. Simulation results demonstrate that the practical case required almost 2 h longer when compared to the baseline case to accomplish the said evacuation. Although the absolute percentage and evacuation duration must be treated carefully, because they are highly reliant on simulation parameters used (e.g., speed, capacity, and network configuration), differences among listed evacuation scenarios are substantial, and relative differences among cases are broadly applicable. After introducing shared mobility into evacuation operation, the performance was observed to have substantially improved in terms of the successfully evacuated population, especially within the first few hours. Accordingly, 50% of the evacuees had successfully left the endangered area within the first 3.75 h of commencement of the evacuation

operation. In the practical case, this level of evacuation required an additional hour. This finding is significant with regard to evacuations with very short deadlines, since undertaking intermediate trips implies that the network output is inefficient during initial stages of the evacuation. The ideal shared-mobility case assumes 100% compliance of evacuee behavior and it sets no constraints on shared mobility usage; thus, the total number of evacuees was greater when compared to that in the general shared-mobility case, because more vehicles were dispatched to the danger area. This represents the theoretical maximum evacuation rate that can be realized with maximum compensation of the trip time via the use of shared mobility. However, evacuation performance in even the ideal shared-mobility case is worse compered to the case sans intermediate trips, thereby highlighting the effects of intermediate trips on the estimated clearance time.

In summary, the simulation results highlight the effects of intermediate trips and shared mobility on the overall evacuation operation. The observed results match our intuition as well as findings of previous researches whilst demonstrating consistency with the questionnaire and interview data.

## 6. Conclusions

The emerging prominence of the shared-economy concept, especially with regard to the idea of shared mobility, is changing the transportation behavior of citizens in modern cities. Shared-mobility service providers organize the spare capacity of vehicles so as to serve the public during their day-to-day commute. The potential of shared mobility in reducing intermediate trips and providing service to vulnerable evacuees has been discussed by investigating public acceptance, driver concerns, and expert opinions, and the outcomes have been validated via results that were obtained from numerical simulations. In accordance with the expert perspective, establishing PPPs between public agencies and shared-mobility companies is an effective way of guaranteeing good performance of shared-mobility services during emergency evacuations. Although shared mobility might not currently be an effective mode of transport for certain vulnerable populations, it has significant potential to become an efficient evacuation mode by reducing intermediate trips and identifying the locations of vulnerable individuals to be evacuated. The major benefits that are associated with the incorporation of shared mobility into the emergency-response scheme include its effectiveness in decreasing the total trip time of the evacuation network by reducing the number of intermediate trips and helping EMAs to identify locations of vulnerable evacuees via one-to-one communication between evacuees and drivers. To overcome potential problems associated with the use of shared mobility during emergency response, a set of implementable strategies has been recommended concerning public agencies and shared-mobility stakeholders based on the expert perspective. PPP represents a sustainable means of developing shared-mobility resources in future. Public agencies must always be ready to leverage new resources and strategies for evacuation planning and operations.

As a future endeavor, the author's intend to develop a model to calculate the economic benefits that are associated with the use of shared-mobility strategies as compared to currently employed evacuation operations.

**Author Contributions:** Conceptualization, M.L.; methodology, X.L.; formal analysis, M.L. and Z.D.; investigation, M.L. and X.L.; writing—original draft preparation, M.L.; writing—review and editing, J.X; project administration, J.X.; funding acquisition, J.X and C.S.

**Funding:** This research was funded by the National Key Research and Development Program of China (Grant no. 2016YFC0802208), the National Natural Science Foundation of China (Grant no. 71101185), and the Natural Science Foundation of Shaanxi Province (Grant no. 2017JQ5122).

**Acknowledgments:** This research was made possible through the cooperation of questionnaire respondents and shared-mobility drivers, and the openness and kindness of interviewed experts in giving their time and opinions.

**Conflicts of Interest:** The authors declare no conflict of interests.

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
